# How to Maintain Safety and Maximize the Efficacy of Cardiopulmonary Resuscitation in COVID-19 Patients: Insights from the Recent Guidelines

**DOI:** 10.3390/jcm10235667

**Published:** 2021-11-30

**Authors:** Dominika Chojecka, Jakub Pytlos, Mateusz Zawadka, Paweł Andruszkiewicz, Łukasz Szarpak, Tomasz Dzieciątkowski, Miłosz Jarosław Jaguszewski, Krzysztof Jerzy Filipiak, Aleksandra Gąsecka

**Affiliations:** 11st Chair and Department of Cardiology, Medical University of Warsaw, 02-091 Warsaw, Poland; dominikachojeckaa@gmail.com (D.C.); jakubpyteocen2@gmail.com (J.P.); 22nd Department of Anesthesia and Intensive Care, Medical University of Warsaw, 02-091 Warsaw, Poland; mateusz.zawadka@wum.edu.pl (M.Z.); pawel.andruszkiewicz@wum.edu.pl (P.A.); 3Department of Clinical Sciences, Maria Sklodowska-Curie Bialystok Oncology Center, 15-027 Bialystok, Poland; lukasz.szarpak@gmail.com; 4Department of Clinical Sciences, Maria Sklodowska-Curie Medical Academy in Warsaw, 00-136 Warsaw, Poland; krzysztof.filipiak@uczelniamedyczna.com.pl; 5Chair and Department of Medical Microbiology, Medical University of Warsaw, 02-091 Warsaw, Poland; tomasz.dzieciatkowski@wum.edu.pl; 61st Department of Cardiology, Medical University of Gdańsk, 80-210 Gdansk, Poland; mjaguszewski@gumed.edu.pl

**Keywords:** COVID-19, SARS-CoV-2, cardiopulmonary resuscitation, CPR, cardiac arrest, OHCA, IHCA

## Abstract

Since December 2019, the novel coronavirus disease 2019 (COVID-19) caused by Severe Acute Respiratory Syndrome Coronavirus 2 (SARS-CoV-2) has remained a challenge for governments and healthcare systems all around the globe. SARS-CoV-2 infection is associated with increased rates of hospital admissions and significant mortality. The pandemic increased the rate of cardiac arrest and the need for cardiopulmonary resuscitation (CPR). COVID-19, with its pathophysiology and detrimental effects on healthcare, influenced the profile of patients suffering from cardiac arrest, as well as the conditions of performing CPR. To ensure both the safety of medical personnel and the CPR efficacy for patients, resuscitation societies have published modified guidelines addressing the specific reality of the COVID-19 pandemic. In this review, we briefly describe the transmission and pathophysiology of COVID-19, present the challenges of CPR in SARS-CoV-2-infected patients, summarize the current recommendations regarding the algorithms of basic life support (BLS), advanced life support (ALS) and pediatric life support, and discuss other aspects of CPR in COVID-19 patients, which potentially affect the risk-to-benefit ratio of medical procedures and therefore should be considered while formulating further recommendations.

## 1. Introduction

The coronavirus disease 2019 (COVID-19) has caused an abrupt growth in the number of hospital admissions due to pneumonia of varying severity. The disease is due to an infection with novel Severe Acute Respiratory Syndrome Coronavirus 2 (SARS-CoV-2), a pathogen first reported in December 2019 [1]. According to the World Health Organization (WHO) at the beginning of October 2021, the number of confirmed cases has reached 223 million, with 4.7 million being fatal [2]. Since the WHO announcement on 11 March 2020, the disease has been declared a pandemic.

COVID-19 disease had a notable impact on the profile of patients requiring cardiopulmonary resuscitation (CPR). According to a single-centered, retrospective, observational study based in Wuhan, China, more than 87% of in-hospital cardiac arrests (IHCA) in COVID-19 patients occurred due to respiratory mechanisms [3]. Moreover, a Swedish national registry-based study estimated the prevalence of shockable rhythms in patients positive for SARS-CoV-2 to be 8%, compared to 23% in SARS-CoV-2-negative cases [4]. Notably, prior to the pandemic, shockable rhythms were the dominant mechanism of out-of-hospital (OHCA) cardiac arrest in adults [5]. The COVID-19 pandemic also raised concerns about the safety of healthcare workers performing medical procedures, including CPR.

In the face of these alterations, the need emerged to optimize current CPR procedures to ensure both the safety of medical personnel, as well as the CPR efficacy for patients. To achieve this, the American Heart Association (AHA), European Resuscitation Council (ERC), International Liaison Committee on Resuscitation (ILCOR) and other resuscitation societies have published modified guidelines that reflect the challenges of the COVID-19 pandemic.

This review summarizes (1) the transmission and pathophysiology of COVID-19, describes (2) the difficulties of CPR in COVID-19 patients, (3) the major changes between the previous and the new recommendations, including modifications of the basic life support (BLS), advanced life support (ALS), and the life support of children and newborns algorithms, and (4) presents further questions and dilemmas regarding CPR in the COVID-19 era.

## 2. Transmission and Pathophysiology

The SARS-CoV-2 infection is being transmitted through several mechanisms. The dominant transmission route is through infected respiratory or oral excretions [6]. Firstly, the virus disseminates within the mucosalivary droplets (size > 5 µm) generated during coughing, sneezing, talking or singing, which come into direct contact with unprotected mucosal surfaces of the mouth, nose or eyes, or are inhaled by people in close proximity (<1 m) to the infected person. Secondly, SARS-CoV-2 spreads through aerosols (or droplet nuclei), i.e., smaller particles (<5 µm) expelled from the respiratory tract, or arising as larger droplets evaporate (airborne transmission) [7]. Larger droplets, when inhaled, tend to settle within the upper portions of the respiratory tract from where they can be effectively eliminated by the ciliary epithelium, whereas aerosols are able to penetrate deeply into the lower respiratory tract, into the alveoli [8,9]. Notably, droplet nuclei may travel as far as 7–8 m away from the host and remain suspended in the air for hours [10], potentially posing a prolonged infection risk.

Other transmission routes of SARS-CoV-2 include direct contact with objects contaminated with infected excretions (fomite transmission) and, rarely, vertical transmission during pregnancy [11]. Additionally, fecal–oral, sexual and bloodborne transmissions are being considered plausible; however, there has been no conclusive documentation of such cases to date.

An angiotensin-converting enzyme (ACE-2), which is highly expressed on the epithelial cells in the pulmonary system, is a functional receptor for SARS-CoV-2 [12]. The primary targets for the virus are the ciliated secretory cells in the nasal epithelium, which allow for viral replication and local propagation, which is accompanied by a local immune response. While in most patients the disease seems to be limited to this stage, in 20% of patients, the pathogens migrate to the lower respiratory tract. There, again through the ACE-2 receptor, the virus infects the type II pulmonary alveolar epithelial cells, leading to their apoptosis and diffuse alveolar damage, as well as a cytokine storm and a systemic inflammatory response [13]. Clinically, alveolar damage results in acute respiratory distress syndrome (ARDS), a life-threatening condition that debilitates gas exchange, resulting in severe hypoxia and consequent cardiac arrest.

Although the pathophysiology of the infection would primarily suggest pulmonary problems, the systemic inflammatory reaction and distribution of ACE-2 receptors in endothelial cells are also associated with cardiovascular disorders [14,15,16]. Cardiovascular disorders in patients with COVID-19 include arterial and venous thrombosis, pulmonary embolism, arrhythmias and myocarditis, as well as acute coronary syndromes and chronic heart failure [17,18,19,20], which may ultimately lead to cardiogenic shock and cardiac arrest in the acute setting and increased cardiovascular morbidity in the chronic setting. Notably, cardiovascular comorbidities are fairly common in COVID-19 patients. It was reported that more than 74% of patients with COVID-19 showed electrocardiogram abnormalities, while cardiac injury was reported in 19.7% of hospitalized patients, being an independent risk factor for in-hospital mortality [14,21]. Cardiac injury in patients with COVID-19 has been associated with a higher demand for non-invasive mechanical ventilation (46.3% vs. 3.9%) or invasive mechanical ventilation (22.0% vs. 4.2%) [21]. Therefore, both pulmonary and cardiovascular complications are responsible for COVID-19-related mortality.

## 3. CPR in COVID-19 Patients

The danger of COVID-19 transmission initiated an intense discourse regarding the balance between the safety of healthcare workers and the patient’s best interests during the pandemic. Any delay to or complication of CPR caused by safety precautions is considered controversial and distressing by many healthcare professionals [22,23]. However, it is the safety of medical service providers that ensures the continuity of healthcare and, therefore, it is considered a high priority.

### 3.1. Initiation of CPR

Direct face to face contact enables COVID-19 transmission, and therefore a policy of social distancing has been implemented worldwide to limit this risk. This change has serious repercussions for patients with cardiac arrest [24], since CPR requires direct proximity to a potentially infected person.

A French registry-based study, which compared over 2500 OHCA before and during the SARS-CoV-2 pandemic, reported that CPR was initiated significantly less frequently both by bystanders (49.8% vs. 54.9%, −5.1 percentage points (95% CI, −9.1 to −1.2)) and mobile medical teams (67.3% vs. 75%, −7.7 percentage points (95% CI, −11.8 to −4.6)) during the pandemic, compared to the pre-pandemic period [24]. Additionally, the same study found the rates of return of spontaneous circulation (ROSC) and 30-day survival to be significantly lower during the COVID-19 period. These findings were confirmed by a systematic review of six papers investigating direct and indirect effects of the pandemic on OHCA outcomes, which concluded that the short-term outcomes of OHCA during the pandemic were worse, with delayed ambulance response and a decreased number of CPR attempts made by bystanders and emergency medical teams [25].

However, another recent systematic review and meta-analysis of five studies comprising the data on 4210 patients found the rate of bystander CPR to be comparable between the groups of COVID-19 suspected or diagnosed patients and COVID-19 not suspected or diagnosed patients (OR = 0.88; 95% CI: 0.63, 1.22; *p* = 0.43) [26]. The same paper confirmed that the rate of survival to hospital discharge was significantly lower among COVID-19 patients compared to the non-COVID-19 group (OR = 0.25; 95% CI: 0.12, 0.53; *p* < 0.001), but the authors explained this finding by the lower incidence of shockable rhythms among COVID-19 patients (OR = 0.19; 95% CI: 0.04, 0.96; *p* = 0.04; I2 = 95%) rather than the unwillingness of bystanders to initiate CPR.

### 3.2. Transmission Risk during CPR

The greatest epidemiologic concern regarding CPR is the risk of airborne transmission of SARS-CoV-2. CPR is an aerosol-generating procedure since it includes several individual procedures that cause aerosolization of the patient’s excretions (i.e., intubation and extubation, manual ventilation, airway suctioning, tracheostomy and non-invasive ventilation) [27].

Estimation of the risk of airborne transmission of SARS-CoV-2 during CPR remains challenging. Conclusive data on the exact risk associated with separate maneuvers are scarce, as CPR procedures are performed almost simultaneously and therefore it is difficult to determine statistically significant differences between them. Based on the experiences and research from the previous coronavirus epidemic (the SARS epidemic, 2002–2004), a Canadian systematic review of five case-control and five retrospective cohort studies investigated the risk of acute respiratory infection transmission during aerosol generating procedures (AGP). The authors classified chest compressions, defibrillation and several maneuvers on the respiratory tract as procedures with increased infectious risk to healthcare workers [28]. Tracheal intubation was associated with the highest risk of transmission (OR 6.6; 95% CI: 2.3, 18.9).

The classification of procedures included in CPR as AGPs has serious implications. It determines the recommended level of personal protective equipment (PPE) for healthcare workers and therefore, in some settings, may cause a several-minute delay to the initiation of CPR, which may potentially impact the patient’s chances of survival. Additionally, PPE affects the quality of CPR and the performance of the CPR team, as it increases the amount of time and energy required to perform CPR. Providing high-quality CPR wearing PPE is a challenge and a test of performance for healthcare providers. Therefore, it is crucial to train the CPR teams to provide the same degree of CPR efficacy to critically ill patients, regardless of whether PPE is used or not.

A recent systematic review of eleven papers by ILCOR, which attempted to estimate the potential risk of transmission associated with chest compressions, defibrillation and CPR in general, concluded that it remains uncertain whether chest compressions or defibrillation generate aerosols and therefore pose a significant risk to medical providers [29]. The authors argued that there is a need to balance the well-known risk of treatment delay in cardiac arrest with the potential risk of infection transmission to the rescuers. Moreover, it was suggested that the time spent in proximity to the infected patient may be a strong determinant of the infection risk, and therefore should be considered before the type of procedure performed [30].

Nonetheless, several pathophysiological mechanisms and organizational aspects of providing CPR (e.g., tidal volumes generated during chest compressions with open–close airway cycling, close proximity to the patient and co-workers, high-stress setting favoring technical mistakes) support the need for special safety precautions during CPR in SARS-CoV-2-infected patients.

### 3.3. Personal Protective Equipment

The multi-route transmission of SARS-CoV-2 dictates the need for introducing various safety measures to avoid infection and limit the spread of the pandemic. One of these measures is the appropriate and rational use of PPE, depending on the estimated transmission risk. During routine, direct care of patients with confirmed or suspected SARS-CoV-2 infection, it is recommended that healthcare workers don droplet-precaution PPE that is sufficient to protect them from large mucosalivary droplets and contaminated surfaces in the patient’s environment [31]. Aerosolization of patients’ excretions during aerosol generating procedures such as CPR and airway management requires a higher level of protection, i.e., donning airborne-precaution PPE. Sets of droplet- and airborne-precaution PPE recommended by the ERC are presented in Figure 1.

Interestingly, it has been highlighted that the fold-type N95 filtering facepiece respirators have been shown to have a significantly greater protection rate for medical professionals during the performance of CPR compared to the cup-type and valve-type masks [33].

As the number of COVID-19 cases mounted worldwide, many countries faced a critical problem of PPE shortages. The situation was particularly challenging during the first months of the pandemic, with disruptions to global supply chains, supply–demand imbalance and inappropriate use of available PPE [34]. In a British cross-sectional observational study, which included over 6000 healthcare professionals, the frequency of contact with suspected or confirmed COVID-19 cases without appropriate PPE was identified as the strongest risk factor associated with the laboratory-confirmed SARS-CoV-2 infection, self-isolation or hospitalization due to suspected or confirmed COVID-19 [35]. In the same study, as much as 22.5% of healthcare workers reported being in a situation without access to certain items of PPE during clinical care of the suspected or confirmed COVID-19 patients. These findings demonstrate that PPE shortages put healthcare personnel at increased risk of SARS-CoV-2 infection, but also at increased risk of temporary obligatory isolation, which is an additional burden for healthcare systems.

The WHO addressed the problem of PPE shortages by issuing interim guidance on the rational use of PPE during the COVID-19 pandemic [36]. The document included strategies for situations when PPE shortages are forecasted, as well as when healthcare facilities are already experiencing limited access to PPE. The specific WHO recommendations are presented in Table 1.

### 3.4. CPR Outcomes in the COVID-19 Era

COVID-19 has largely influenced the outcomes of CPR, as the harmful effects of infection will have an influence on the CPR efficacy. In the case of OHCA, in a US registry-based study, ROSC was achieved in 29.8% of patients in 2019 and in only 23.0% patients in 2020 (adjusted RR = 0.82; 95% CI: 0.78, 0.87; *p* < 0.001) [37]. The difference was even more pronounced for IHCA patients, who in 2019 achieved the ROSC rate in 56% of cases, which decreased to 36% during the COVID-19 pandemic, according to a single-center study investigating nearly 250 IHCA cases [38]. In a Swedish national registry-based study, patients with confirmed ongoing COVID-19 infections had significantly lower rates of successful ROSC following IHCA (30.6%) [4] compared to patients without COVID-19, with the reported rates being lower for patients with COVID-19 pneumonia (13.2%) [3]. Interestingly, non-COVID-19 patients had higher ROSC rates in 2020 than before the pandemic (52.6 vs. 47.2%) [4], which might be due to the increased alertness of medical personnel.

However, a recent systematic review and meta-analysis of four studies comparing IHCA before and during the COVID-19 pandemic concluded that the IHCA outcomes in pre-COVID-19 and COVID-19 periods were statistically similar in terms of survival to hospital discharge (35.6% vs. 32.1%, OR = 1.72; 95% CI: 0.81, 3.65; *p* = 0.16), ROSC (51.9% vs. 48.7%, OR = 1.27; 95% CI: 0.78, 2.07; *p* = 0.33), cardiac arrest recurrence (24.9% vs. 17.9%, OR = 1.60; 95% CI: 0.99, 2.57; *p* = 0.06) and overall mortality (65.9% and 67.2%, OR 0.67; 95% CI: 0.33, 1.34; *p* = 0.25) [39]. However, the results should be interpreted with caution due to the high heterogeneity between the studies (30–70%).

The concern about the impact of the COVID-19 pandemic on cardiac arrest outcomes is well justified and further research is needed to improve the understanding of this issue.

## 4. Basic Life Support in COVID-19

As the rates of SARS-CoV-2 infection vary across countries, the risks of transmission during BLS procedures vary as well. To reduce the spread of infectious agents, all patients should be treated with precautions as potential carriers [40], yet the method of resuscitation must remain productive, with no loss in BLS efficiency. For BLS in adults with suspected or confirmed COVID-19, the recommendations show a clear distinction between procedures with available healthcare personnel, emergency medical dispatch staff and non-medical rescuers. A summary of novel BLS recommendations for adult COVID-19 patients is presented in Figure 2.

### 4.1. BLS Recommendations for Healthcare Personnel

Guidelines for BLS by healthcare workers in adults with suspected or confirmed COVID-19 clearly state that the availability of airborne-precaution PPE and their proper use is crucial. Thus, the BLS teams should only include healthcare workers with proper training to use airborne-precaution PPE [32]. The airborne-precaution PPE should always be used for procedures that might generate aerosols, such as chest compressions, airway and ventilation interventions [32]. The personnel should be limited to a minimum, allowing for efficient BLS, but limiting potential infections [41].

Cardiac arrest should be recognized primarily by visual evidence. Healthcare workers should predominantly look for signs of missing or abnormal breathing and the general absence of indications of life. The AHA recommends that checking for a pulse should be included in the standard condition assessment, yet the ERC and ILCOR do not underline the importance of such a routine [32,41].

If no normal breathing and/or pulse is found, cycles of 30 compressions and 2 breaths should be performed [32]. AHA guidelines recommend that manual chest compressions should be replaced with mechanical CPR devices if available, to reduce the number of required rescuers, thus lowering the risk of infection spread [41]. Additionally, to minimize the risk of aerosol generation, chest compressions should be paused during ventilations. Breaths should be provided with a bag-mask device with a high-efficiency particulate air filter (HEPA) or a heat and moisture exchanger filter (HME). To ensure the tight seal for bag-mask ventilation, two hands should be used to hold it against the patient’s face, thus requiring a second person in the team [32]. Notably, if such a mask is not available or the BLS team is unprepared for its administration, the ERC recommends abandoning breaths in favor of passive oxygenation using an oxygen mask with provided oxygen [32]. Earlier guidelines advertised the usage of a supraglottic airway during the course of resuscitation, yet more recent publications suggest that it increases the chance of aerosol generation and therefore should be omitted [42].

A defibrillator or an automated external defibrillator should be used if available. The application of defibrillator pads and delivering a shock from the device is not considered to be an aerosol-generating procedure, thus it can be safely executed by the rescuer wearing only droplet-precaution PPE [32]. It is vital that any rescuers are informed of the COVID-19 status of the patient before entering the resuscitation site [41].

### 4.2. BLS Recommendations for Emergency Medical Dispatch Staff

Recommendations for emergency medical dispatch staff advise that the teams of first responders or volunteers should only be dispatched if they received training in the use of PPE and have proper access to that equipment [32]. On the site of an emergency, rescuers should quickly evaluate the risk of COVID-19 infection based on the encountered situation and alert the healthcare personnel if there is an infection hazard.

For untrained rescuers, rescue breaths should be omitted entirely as the guidelines advise the compression-only protocol [32,40]. Additional rescuers or bystanders should be guided to the nearest automated external defibrillator (AED), if available [32]. The ERC guidelines underline that if first responders or volunteers have only droplet-precaution PPE, for patients with suspected or confirmed COVID-19 infection, the resuscitation should only include the use of an AED. In that case, chest compressions or rescue breaths are not recommended, as they would put the emergency medical dispatch staff at risk of becoming infected during the procedure [32].

### 4.3. BLS for Lay Rescuers

As the proper PPE is usually unavailable for lay rescuers, the guidelines focus on decreasing the risk of infection with simple and widely available tools. After assessing the safety of rescue site, responsiveness should be evaluated by shaking the victim and shouting. Rescuers should examine the breathing patterns without opening the airways of the sufferer and avoid approaching the nose and mouth to an extent that would not impair their judgment. If the person does not breathe normally and is unresponsive to verbal and physical stimuli, the ERC recommends that a cardiac arrest should be identified [32]. If the COVID-19 status of the victim is known, the emergency medical services should be notified about it immediately.

Lay rescuers should follow the compression-only protocol of resuscitation [42,43]. It is recommended to cover the person’s nose and mouth with a face mask or a cloth in order to reduce the aerosol and droplet generation during chest compressions [41]. An AED should be used if available [41,43]. Rescuers should thoroughly wash and disinfect their hands promptly after the BLS procedures [40]. It is important that the medical personnel are contacted in order to screen the rescuers for COVID-19 after having contact with a person with a confirmed or suspected infection [32].

## 5. Advanced Life Support in COVID-19

The modified advanced life support (ALS) guidelines for COVID-19 patients proposed by scientific societies mainly focus on ensuring the safety of healthcare workers by setting clear priorities: first—self, second—colleagues and third—the patient [32]. This goal should be achieved through improvements in three areas: limiting the potential exposure of healthcare workers, optimal and safe advanced airway management and assessment of appropriateness of CPR [41]. A summary of novel ALS recommendations for adult COVID-19 patients is presented in Figure 3.

### 5.1. Limiting Exposure during CPR

ALS interventions are a natural continuation and escalation of BLS and therefore follow the same general safety principles. Healthcare workers should identify cardiac arrest solely based on the patient’s unresponsiveness and lack of normal breathing, which are to be assessed from a safe distance without approaching the patient’s face [32]. A call for help or the arrival of any new providers to the patient’s room should be accompanied by clear communication of the patient’s COVID-19 status to make all rescuers aware of the potential risk of transmission [41]. As mentioned before, it is advised to limit the number of personnel in the room to the indispensable minimum and to close the door to prevent the potential spread of infectious aerosols [41].

Most importantly, the modified guidelines by the ERC and the AHA unanimously recommend that CPR should only be performed in full airborne-precaution PPE. The only action that may be attempted wearing only droplet-precaution PPE is defibrillation, which is not considered a significant source of aerosol generation [44]. The ERC recommends applying up to three shocks for ventricular fibrillation or pulseless ventricular tachycardia while colleagues are donning full airborne-PPE to start chest compressions [32].

Additionally, the AHA encourages early implementation of mechanical CPR devices for suitable patients in order to further decrease the number of rescuers and their proximity to the patient [41]. The ERC only advises the use of mechanical CPR devices for prolonged CPR [32]. Notably, mechanical CPR devices are also beneficial to the patient, since wearing PPE and the consequent fatigue or discomfort of the rescuers may compromise the quality of chest compressions [45].

### 5.2. Airway Management during CPR

The new guidelines put great emphasis on proper airway management in COVID-19 patients. Airway maneuvers pose the highest infection risk to healthcare workers performing CPR. However, they address the pathophysiological background of COVID-19, i.e., ARDS and hypoxemia, and therefore enable causative treatment of cardiac arrest.

The guidelines agree that intubation with a cuffed endotracheal tube connected to a HEPA or HME filter facilitates effective ventilation with the lowest risk of aerosolization, and therefore must be prioritized. To maximize the chances for first-pass success, intubation should be performed by the most experienced and skilled provider and with minimal interruptions, i.e., without simultaneous chest compressions. The guidelines also suggest using video laryngoscopy to increase the distance between the provider and the patient’s respiratory tract, but it may also improve the efficacy of intubation, especially in patients with a difficult airway [46]. Notably, the AHA advises to avoid emergency intubations in COVID-19 patients [41]. This recommendation corresponds with guidelines issued by British anesthetist societies, which state that emergency intubation poses an increased risk of disease transmission due to a rushed, high-stress setting and lack of a pre-defined and well-prepared airway strategy [47]. Patients who are already intubated at the time of cardiac arrest should be left on mechanical ventilation, although the ventilator settings need to be adjusted to enable safe and effective asynchronous ventilation [32,41].

When swift intubation is not available, manual ventilation with a supraglottic device connected to a HEPA or HME filter should be attempted. Although second-generation supraglottic airway devices facilitate significant air leakage and aerosol spread [48], they are considered superior to manual bag-mask ventilation. However, healthcare providers are also advised to only use well-trained techniques that they feel confident performing [47]. Therefore, the decisions regarding ventilation methods should always be dictated by the current capabilities of the personnel and the healthcare facility.

Patients with COVID-19 are often nursed in prone position to improve ventilation-perfusion matching in the lungs and increase blood oxygenation. Those patients may be unconscious and already intubated, or awake with no advanced airway. According to the modified guidelines by the ERC and the AHA, CPR may be delivered in a prone position in intubated patients [32,41]. However, it is advisable to turn the patient supine if there are concerns regarding the effectiveness of chest compressions, airway management or if ROSC is not obtained within minutes [32]. In prone-positioned, un-intubated patients, the guidelines recommend to attempt turning the patient supine in the first place, before starting chest compressions [32,41]. However, the process of turning the patient supine may prove time-consuming and challenging with several healthcare workers to be engaged and numerous potential adverse events (e.g., displacements and disconnections of medical devices). This may result in additional exposure and increased transmission risk for healthcare workers as well as time loss for the patient [49]. The available literature supports the effectiveness of prone CPR [50] and therefore it is reasonable to consider it as a temporary measure if an immediate turn into the supine position is not available.

### 5.3. Appropriateness of CPR

Another crucial aspect of providing critical care is the prevention of forecasted cardiac arrest as well as an assessment of the appropriateness of CPR. Both the ERC and the AHA strongly recommend advanced care planning [32,41]. This includes early identification of patients who are at risk of serious deterioration with physiological track-and-trigger systems (e.g., NEWS2 recommended by the Resuscitation Council UK) [51] that enable timely care escalation and implementation of proactive safety precautions (e.g., optimal airway management, transferring the patient to a negative pressure room) [41]. Additionally, clinicians should attempt to discuss the goals of care preferred by the patient or their proxy in order to avoid unwanted treatment. Poor outcomes of CPR in certain groups of critically ill COVID-19 patients, as well as the pandemic-related resource scarcity, should also be taken into consideration when making decisions about attempting CPR. In this setting, the proper use of do-not-resuscitate orders is crucial, as it enables the fair and responsible allocation of scarce medical resources as well as providing a respectful approach to the patient’s goals and values [52].

## 6. Basic and Advanced Life Support of Children and Newborns

The onset of COVID-19 in children is considered to be mild, as in a Chinese case series of 2135 pediatric patients, no more than 6% developed severe or critical illness [41]. This is very likely to result in the underdiagnosis of COVID-19 among children and, thus, an underestimation of the infectious potential of pediatric patients. As children mainly require resuscitation due to respiratory, not cardiovascular, mechanisms of cardiac arrest, even an unrecognized infection should be treated as a possible etiology, particularly in children with comorbidities and neonates, who have been observed to be more prone to dangerous forms of SARS-CoV-2 infection [53]. It is crucial to remember that although pediatric patients are more likely to require resuscitation as a consequence of other etiologies, they might still pose a risk for medical personnel or any bystanders due to an unrecognized infection.

In the case of children and newborns, there is an observed tendency to undermine the safety precautions by rescuers, both bystanders and healthcare workers. This might be connected to the evaluation of rescuers’ personal risks as being less important than the potential benefits for the child, particularly if the bystanders are caregivers or household members of the victim. The recommendations point out that the child’s household members are likely to have already been exposed to the virus; nonetheless, it should be underlined that the precautions mentioned in this work are crucial not only for protecting the rescuers’ own lives and health, but also the lives and health of their whole community [32]. A summary of novel life support recommendations for pediatric patients is presented in Figure 4.

### 6.1. Pediatric Life Support

The protection of rescuers is an important problem in the guidelines aimed at pediatric patients as well. As children are likely to be asymptomatic in terms of COVID-19, it is recommended to treat pediatric patients by default as being infected with SARS-CoV-2, and thus taking into consideration a contagion risk for the resuscitators [40]. The donning of PPE by healthcare providers is recommended when treating a critically ill child who has confirmed or suspected COVID-19 [54]. The ERC suggests that the type of PPE for attending pediatric patients should be chosen proportionately to the presumed risk of SARS-CoV-2 transmission. Lay bystanders should abstain from actions with a high risk of transmission and remain an appropriate distance from the victim [32].

The modified BLS algorithm for children requires assessing their responsiveness and breathing visually, e.g., by observing a chest rise, but the ERC allows for placing a hand on the belly of the victim [32]. Rescue breaths are recommended to be delivered with a bag-mask device with tight seal and a HEPA/HME filter, as for other age groups [41]. The AHA recommends delivering rescue breaths for pediatric patients even if the mask is unavailable, provided that the rescuers are willing and able to perform them [43]. On the contrary, the ERC advises applying compression-only CPR in children in case of the unavailability of a proper bag-mask device, and placing a surgical mask over the child’s mouth and nose [32]. The ERC guidelines also discourage the use of a cloth as an alternative, as the risk of airway obstruction or substantial restriction of passive air movement is a major factor for pediatric patients [32]. An AED, if available, should be used as soon as possible [32,43].

Airway management is crucial for pediatric patients. The airway should be opened and maintained by means of positioning and a head tilt with a chin lift or jaw thrust when performing a bag-mask ventilation [32]. Supplemental oxygen should be used as soon as possible. When using oxygen-providing devices (oxygen mask, nasal cannula or a non-rebreathing mask), a surgical mask should be put on the face of all patients with suspected or confirmed COVID-19. The medication should not be given through a nebulizer, but with a metered-dose inhaler [32].

ALS guidelines for pediatric patients suggest limiting the resuscitation team to four people, who should don airborne-precaution PPE before attempting CPR. Tracheal intubation is a priority and should be performed as soon as possible, preferably using video laryngoscopy [40,54]. The chest compressions, if begun, should be paused during an intubation attempt [32,54]. If intubation is delayed, the AHA recommends considering a supraglottic airway or a bag-mask device with filter and tight seal to improve airway stability [41]. The ventilator should be prepared beforehand, as it is recommended to be used immediately after the intubation. Guidelines suggest an adjustment of the respirator parameters, for instance to a reference proposed by SEMICYUC: FiO2 set to 1.0 with a respiratory rate of 10–12 breaths per minute, pressure-controlled mode with a tidal volume of 6 mL/kg of ideal body weight, adjusted positive end-expiratory pressure (PEEP) and alarms with the trigger turned off [40]. Furthermore, the ERC recommends inserting a viral filter (HEPA or HME) between the breathing circuit and the patient’s airway, as well as an additional filter on the expiratory limb of a ventilator in order to limit aerosol spread. Only using closed suction systems and administering a neuromuscular blocking drug in order to prevent coughing is also advised [32].

When a defibrillator or an analogous device is available and a VF/VT rhythm is identified, a shock should be delivered, followed by chest compressions. If the asystole or PEA non-shockable rhythms are recognized, or the first shock of a patient with VF/VT rhythm is ineffective, the AHA suggests delivering epinephrine every 3–5 min and, if possible, choosing mechanical chest compression devices over compressions by hand [41]. Interestingly, the ERC suggests that if the resuscitation team is not yet wearing airborne-precaution PPE, while other healthcare workers are donning PPE, a rescuer should deliver up to two additional shocks if they are indicated [32].

### 6.2. Neonatal Life Support

Although the vertical transmission of COVID-19 infection is considered rare [55,56] and the risk of becoming infected at birth, even if born to a confirmed positive mother, is low [32], there is a need for recommendations regarding the resuscitation of newborns. It should be underlined that the COVID-19 status of the mother is particularly important in procedures regarding neonates, who are dependent on their mothers and should not be isolated from them if possible. Breast feeding and skin-to-skin care should be allowed, as long as strict hand hygiene is ensured and a fluid resistant surgical mask is worn by the mother to reduce the risk of droplet spread [32]. Providers of resuscitation for newborns should be aware of the mother being a potential source of aerosolization and infection. It is important that the rescuers don appropriate PPE when approaching the child, if the mother has a suspected or confirmed COVID-19 infection.

The AHA guidelines indicate that activities connected with routine neonatal care and the initial steps of resuscitation (e.g., drying, tactile stimulation, placement into a plastic bag or wrap, assessment of heart rate, and placement of pulse oximetry and electrocardiographic leads) are unlikely to be aerosol generating. The same recommendations point towards the endotracheal instillation of medications and suction of the airway after delivery as the AGPs and thus endorse not performing the suction routinely for clear or meconium-stained amniotic fluid, and refraining from endotracheal instillation altogether [41]. Staff performing the basic and advanced life support where maternal COVID-19 is suspected or confirmed are recommended to don full airborne-precaution PPE [32].

The general approach for resuscitation of newborns has not been majorly changed by any of the societies. The safety precautions and airway management mirror the modifications recommended for other pediatric patients.

## 7. Caution, Questions, Dilemmas

The SARS-CoV-2 pandemic has been around for almost two years now, and so has the research on this newly emerged issue. Naturally, many of the currently available studies on COVID-19 are based on the data acquired during the first months of the pandemic. The modified CPR guidelines were issued in June 2020 by both the ERC and the AHA in response to the initial challenges of the pandemic [32,41]. Although numerous knowledge gaps still need to be filled, our understanding of COVID-19 is today undoubtedly better than it was a year ago. Policymakers and healthcare systems worldwide have now partly adjusted to the new reality, and the situation, although still challenging and dangerous (see Figure 5), is not as surprising and as novel as before.

This provokes an interesting question—have we reached the point where the relation between the potential risks and benefits of medical procedures has changed? The steps taken to protect healthcare workers during CPR are well justified. However, there have also been concerns about whether the new recommendations may have been unintentionally misused, compromising patients’ best interests [57]. We can now possibly implement new evidence-based solutions and recommendations to further optimize and guide the functioning of healthcare during the SARS-CoV-2 pandemic.

### 7.1. Infectivity of COVID-19 Patients

The great infectious potential of SARS-CoV-2 is attributed to its ability to replicate efficiently in the upper respiratory tract and its intense pharyngeal viral shedding at a very early stage of the disease [58]. In fact, in community settings, nearly 50% of infections are caused by pre-symptomatic hosts with an estimated contribution to the basic reproductive number (R0) of 0.9, which alone is almost sufficient to maintain the pandemic [59,60]. The infectiousness of SARS-CoV-2 peaks at symptom onset, but viral shedding is thought to begin 5–6 days earlier [60] and to last for a median of 20 days [61]. Notably, excretions obtained from infected people beyond day 8 since symptom onset, although still positive in real-time polymerase chain reaction (RT-PCR) tests, do not seem to be clinically infectious [62].

When it comes to severe cases of COVID-19, the mean viral load is higher and the shedding period lasts longer compared to mild cases [63,64], increasing the infection risk for healthcare workers. In a systematic review and meta-analysis of 11 studies, it was estimated that during the first four months of the SARS-CoV-2 pandemic, healthcare workers constituted 10.1% (95% CI: 5.30, 14.90) of all COVID-19 cases, although with a milder course and lower mortality than the general population [65]. Although community transmission of SARS-CoV-2 plays a significant role in healthcare workers as well, the occupational risk of contracting the disease should never be undervalued, since performing medical procedures requires direct, close and prolonged contact with infected patients, potentially infected colleagues and contaminated environments.

Another aspect of establishing the infectivity of COVID-19 patients is the correct interpretation of the RT-PCR results. RT-PCR, a standard method for diagnosing SARS-CoV-2 infection, evidences only the presence of viral RNA and not necessarily viable, infectious virions. In a study by Bullard et al., which included 90 RT-PCR SARS-CoV-2-positive samples, the ability to infect culture cells was observed only for specimens obtained < 8 days post symptom onset and with a cycle threshold < 24 [62]. Another study by Basile et al., which analyzed 234 RT-PCR SARS-CoV-2-positive samples, found a cycle threshold > 32 to rule out infectivity [66]. The authors of the study concluded that viral culture may prove clinically useful to guide decisions about de-isolation.

Although further research is needed to determine the usefulness of such an approach for both mild and severe cases of COVID-19, establishing the infectivity status may enable better patient cohorting, more accurate risk assessment by healthcare workers and therefore improved patient care.

### 7.2. Immunity Status of Healthcare Workers

Another issue regarding the assessment of the occupational transmission risk of SARS-CoV-2 is the immunity status of healthcare workers. The longevity of immunity acquired through natural infection remains unknown. However, there are some promising results suggesting that immune memory following the SARS-CoV-2 infection may last for months in most individuals [67]. The experiences from previous coronavirus epidemics also support the possibility of maintaining long-lasting, protective immunity against SARS-CoV-2 [68]. New publications on this issue are eagerly awaited to establish whether the risk of infection may be durably and reliably decreased for COVID-19 convalescents.

Most importantly, it has now been possible to acquire immunity against SARS-CoV-2 through immunization as well. The intense intellectual and organizational effort to combat the pandemic resulted in the development of hundreds of COVID-19 vaccine candidates within just months. Currently, there are three vaccines approved by the Food and Drug Administration (FDA) for use in the US [69] and four vaccines approved by the European Medicines Agency (EMA) for use in the European Union [70]. The efficacy of all vaccines is satisfactory, amounting to 95% for BNT162b2 mRNA [71], 94.1% for mRNA-1273 [72], 70.4% for ChAdOx1 (AZD1222) [73] and 66.1% for single-dose Ad26.COV2.S [74] vaccines. Some countries prioritized healthcare workers in their first dose allocation, a policy recommended by the Centers for Disease Control and Prevention [75]. Notably, when it comes to CPR, any deviation from the standard life support guidelines, reflecting the perceived risk of disease transmission, may become disadvantageous to both COVID-19 and non-COVID-19 patients. The vaccines against COVID-19 successfully prevent hospitalization and the severe course of the disease; however, the protection declines with time [76]. It is now important to assess how much safety was granted to healthcare workers by completing the vaccination against SARS-CoV-2 and whether it should be followed by new modifications to the interim COVID-19 guidelines.

### 7.3. Impact of Personal Protective Equipment on CPR Quality

Prior to the SARS-CoV-2 pandemic, there had already been concern about PPE’s negative impact on the overall performance of healthcare workers and specifically on CPR quality. It was debated that the use of respirators and whole-body gowning interferes with numerous physiological functions of the wearer, as well as psychological and social aspects of work, leading to decreased task performance [77]. PPE was also shown to worsen the quality of chest compressions provided by anesthesia residents in a randomized crossover simulation study [78].

These considerations reappeared quickly as the COVID-19 crisis in healthcare escalated. In 2020, several studies showed that wearing PPE significantly compromised the quality of chest compressions and increased the fatigue of healthcare workers performing CPR [45,79,80]. These findings were confirmed by a systematic review and meta-analysis of publications up to 6th June 2020 [81].

Notably, a more recent triple-crossover simulation trial demonstrated that the use of FFP2 masks by emergency medical technicians did not influence the depth of chest compressions provided during 12-min BLS scenarios, although it did increase subjective physical exhaustion among participants [82]. The authors of the study debated that the ability to provide high-standard chest compressions achieved during emergency medicine training cannot be easily altered by external factors, like wearing PPE. In another recent simulation, a randomized crossover trial by Rauch et al. showed no significant difference between two-minute sequences of chest compressions performed with PPE (including a FFP3 mask) and without PPE in terms of several quality indicators, i.e., depth, rate, release and number of effective compressions [83]. Interestingly, participants of the study also reported increased subjective fatigue (*p* < 0.001) and decreased subjective performance (*p* = 0.031) while wearing PPE, which was not reflected by the objective measurements of chest compression quality. This questions the ability of healthcare providers to accurately self-assess the quality of their performance.

Nonetheless, the findings suggestive of the negative impact of full PPE on CPR quality remain a concern. They support the need to introduce changes to CPR algorithms (e.g., more frequent changes of chest compression providers, broader use of mechanical CPR devices) as well as improvements in the assessment of transmission risk and PPE guidelines in order to secure patients’ chances of survival. Perhaps further high-fidelity simulation trials will ensure more understanding of this issue.

## Figures and Tables

**Figure 1 jcm-10-05667-f001:**
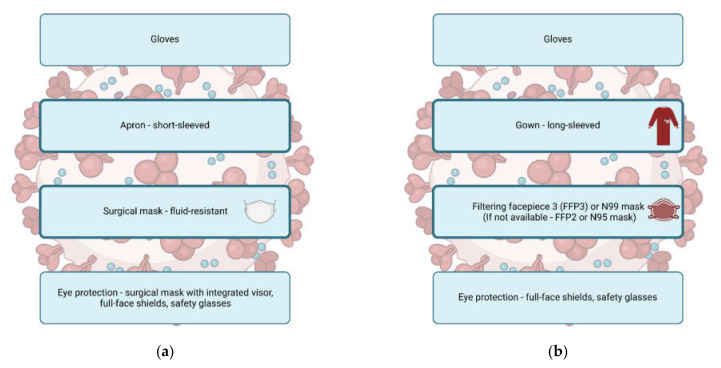
Minimum level of personal protective equipment (PPE) recommended by the European Resuscitation Council [32]. (**a**) Minimum level of droplet-precaution PPE. (**b**) Minimum level of airborne-precaution PPE.

**Figure 2 jcm-10-05667-f002:**
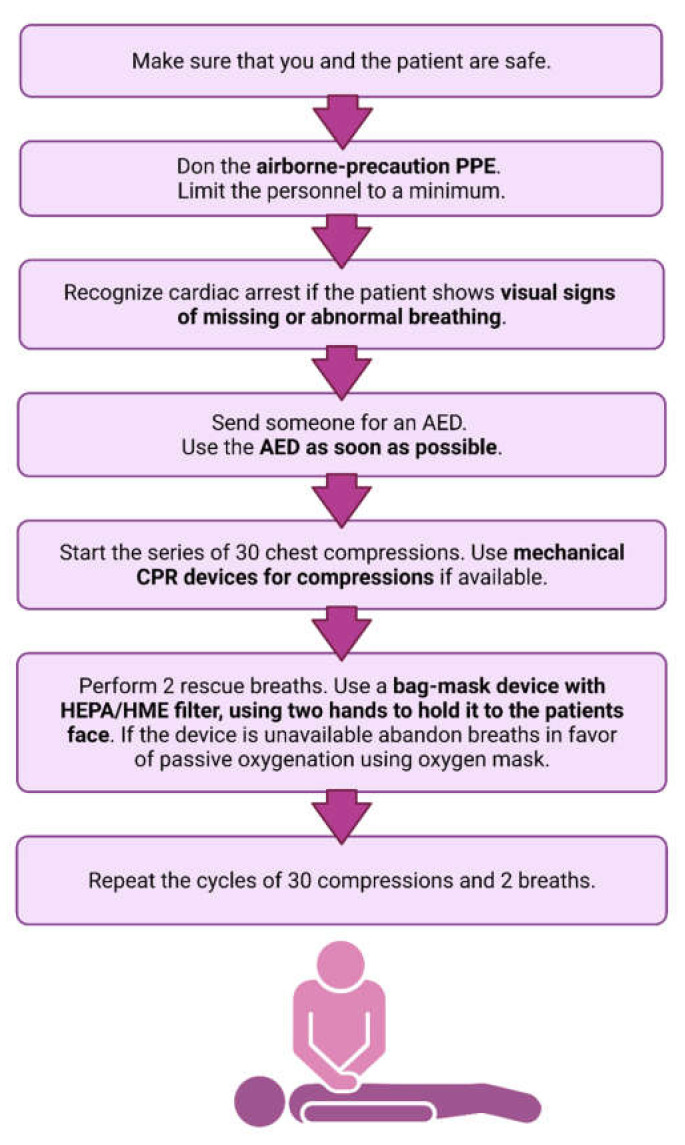
A summary of novel Basic Life Support recommendations. The modified recommendations are marked in bold.

**Figure 3 jcm-10-05667-f003:**
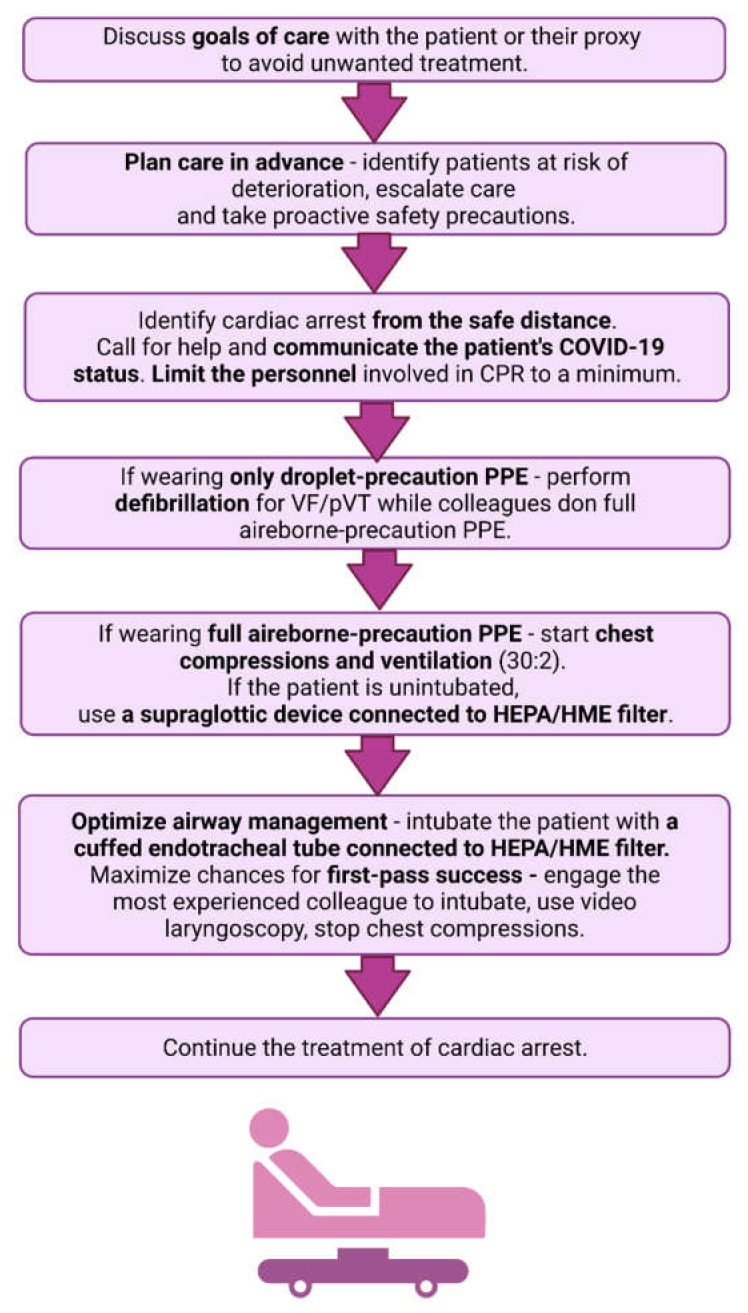
A summary of novel Advanced Life Support recommendations. The modified recommendations are marked in bold.

**Figure 4 jcm-10-05667-f004:**
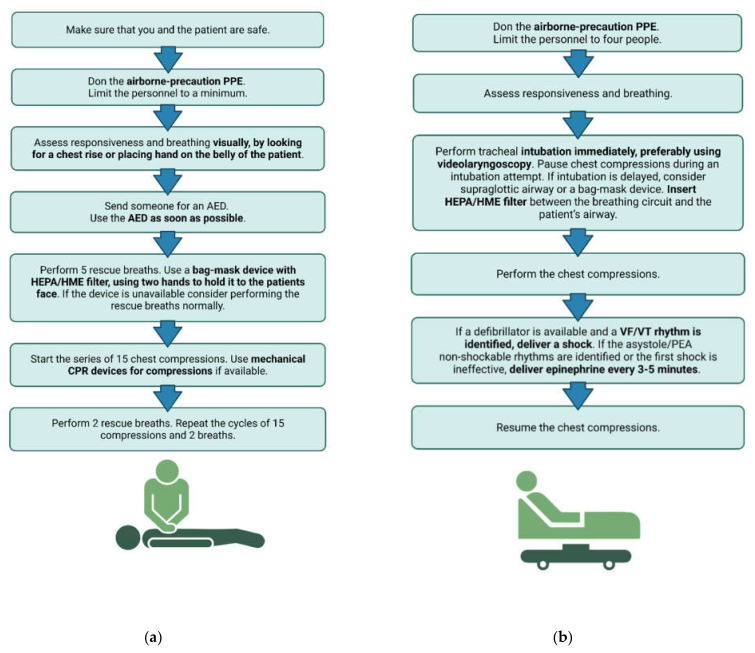
A summary of novel life support recommendations for pediatric patients. The modified recommendations are marked in bold. (**a**) Basic Life Support for pediatric patients. (**b**) Advanced Life Support for pediatric patients.

**Figure 5 jcm-10-05667-f005:**
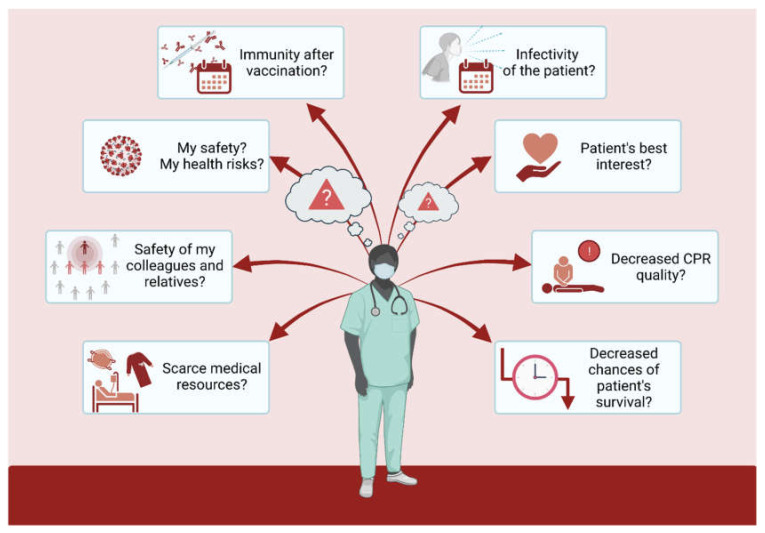
Dilemmas regarding uncertainties around cardiopulmonary resuscitation in times of COVID-19, faced by medical personnel.

**Table 1 jcm-10-05667-t001:** Strategies for personal protective equipment shortages during COVID-19 pandemic recommended by the World Health Organization [36].

Situation	Strategies
**Forecasted shortages of PPE**	**Optimized PPE use** Consider alternative methods of patient–doctor contact, e.g., online consultations, telemedicine. Implement physical barriers between patients and healthcare workers, e.g., glass/plexiglass screens, observational windows, transparent curtains. Cohort COVID-19 patients in the same room with a dedicated team of healthcare workers. Limit the number of healthcare workers entering the rooms of COVID-19 patients as well as the overall frequency of direct contact, e.g., by bundling several case activities. Improve the ability of healthcare workers to perform risk assessment and appropriate selection of PPE. Limit the traffic of visitors in inpatient healthcare facilities.
	**Rational and appropriate PPE use** Apply transmission-based precautions appropriately to the risk of infection and only when patients are still infectious. Avoid using excessive PPE, e.g., shoe protection, head covers, coveralls, double layered gloves and gowns (not required for COVID-19).
	**Coordination of PPE supply chains**
**Severe shortage of PPE**	**Extended use of PPE** Prolonged use of PPE items for several clinical encounters among cohort of confirmed COVID-19 patients without other transmissible infections.
	**Re-use of decontaminated or reprocessed PPE** Extraordinary decontamination or reprocessing of single-use PPE items according to manufacturers’ instructions and local regulations.
	**Alternative PPE** Temporary replacement of appropriate PPE by alternative items, e.g., FFP1 respirators, non-medical fabric masks, launderable or disposable aprons, lab coats, homemade face shields, powered air purifying respirators (PAPR), protective gloves used in other industries.
	**Note:** The proposed strategies for severe shortages of PPE are associated with serious limitations and therefore should be considered only as temporary last resort measures to ensure the continuity of healthcare.

## Data Availability

Data sharing not applicable.

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
