# Peer review of "How to Maintain Safety and Maximize the Efficacy of Cardiopulmonary Resuscitation in COVID-19 Patients: Insights from the Recent Guidelines"

_jcm, 2021, doi:10.3390/jcm10235667_

Round 1
Reviewer 1 Report
The authors have done reviewed regarding basic life support in COVID 19 pandemic. My comments are
- This was a nice review and will benefit for physician and layperson.
- There was need knowledge attitude and practice regarding the covid for healthcare workers and lay person. https://doi.org/10.1177/10105395211011017 ;
- The need for research or further studies regarding the basic life support time for ventilation and perfusion in COVID 19 era.
- How about the time of compression in basic life support, considering the helper using N95, for optimal compression.
Author Response
Dear Reviewer,
we are thankful for the time and effort that you spent to provide in-depth review of our manuscript.
We corrected our manuscript according to your suggestions. Our response and corrections are listed below.
1. This was a nice review and will benefit for physician and layperson.
We would like to thank the Reviewer for appreciating our manuscript.
2. There was need knowledge attitude and practice regarding the covid for healthcare workers and lay person. https://doi.org/10.1177/10105395211011017
3. The need for research or further studies regarding the basic life support time for ventilation and perfusion in COVID 19 era.
4. How about the time of compression in basic life support, considering the helper using N95, for optimal compression.
We thank the Reviewer for this question. Recently published papers point out that the N95 masks greatly influence the fatigue of the helper, which can result in a lower quality of chest compressions. Considering that, we found no recommendations, that would suggest shortening the time of compression beneath 2 minutes for each rescuer. As the resuscitation teams in
times of COVID-19 are recommended to be small, there is no sufficient research to predict if the shortened compression cycles and more frequent rotation of a small group of rescuers will contribute to higher efficacy of cardiopulmonary resuscitation.
Altogether, we are grateful for the in-depth revision of our manuscript and we hope that it will be considered for publication in “Journal of Clinical Medicine”.
On behalf of all Authors,
Sincerely,
Jakub Pytlos and Aleksandra Gasecka
Reviewer 2 Report
As one of the readers, I thank the authors for a good summary of the many findings and known facts.
This paper is already good enough, but to make it more complete, I give some of my opinions.
- There seems to be no precedent for previously published papers and presentations and recent papers on OHCA before the pandemic and after the pandemic. For CPR that will harm and potentially affect you, COVID-1 will affect CPR.
- According to airway management guidelines, there are many aerosols and the use of many barrier devices for protection. I think the Korean author's explanation would be better.
- PPE is beyond the study results for laurels on CPR performance. The PPE advises on the effectiveness of education on the activities of the CPR team based on the advice of critically ill patients. Achieving CPR wearing PPE is an assessment of CPR performance, an estimate of CPR performance, based on the performance of health care providers.
- PPE affects the quality of CPR, and the performance of the CPR team due to time constraints and physician involvement.
Author Response
Dear Reviewer,
we are thankful for the time and effort that you spent to provide in-depth review of our manuscript.
We corrected our manuscript according to your suggestions. Our response and corrections are listed below.
1. There seems to be no precedent for previously published papers and presentations and recent papers on OHCA before the pandemic and after the pandemic. For CPR that will harm and potentially affect you, COVID-1 will affect CPR.
We thank the Reviewer for this comment, which we added to the manuscript as follows: The harmful effects of infection will have an influence on the CPR efficacy.
2. According to airway management guidelines, there are many aerosols and the use of many barrier devices for protection. I think the Korean authors explanation would be better.
We thank the Reviewer for this suggestion. We referred to this interesting work in the discussion and added it to the reference list.
3. PPE is beyond the study results for laurels on CPR performance. The PPE advises on the effectiveness of education on the activities of the CPR team based on the advice of critically ill patients. Achieving CPR wearing PPE is an assessment of CPR performance, an estimate of CPR performance, based on the performance of health care providers.
We thank the Reviewer for this comment, which we added to the manuscript as follows: Providing high quality CPR wearing PPE is a challenge and a test of performance for healthcare providers. Therefore, it is crucial to train the CPR teams to provide the same degree of CPR efficacy to critically ill patients, regardless whether PPE is used or not.
4. PPE affects the quality of CPR, and the performance of the CPR team due to time constraints and physician involvement.
We thank the Reviewer for this comment. We added this information to the manuscript as follows: PPE affects the quality of CPR and the performance of the CPR team, as it increases the amount of time and energy required to perform CPR.
Altogether, we are grateful for the in-depth revision of our manuscript and we hope that it will beconsidered for publication in “Journal of Clinical Medicine”.
On behalf of all Authors,
Sincerely,
Jakub Pytlos and Aleksandra Gasecka